# Mitochondria-Induced Immune Response as a Trigger for Neurodegeneration: A Pathogen from Within

**DOI:** 10.3390/ijms22168523

**Published:** 2021-08-07

**Authors:** Marta Luna-Sánchez, Patrizia Bianchi, Albert Quintana

**Affiliations:** 1Institut de Neurociències, Universitat Autònoma de Barcelona, 08193 Bellaterra, Barcelona, Spain; marta.luna@uab.cat (M.L.-S.); Patrizia.Bianchi@uab.ca (P.B.); 2Department of Cellular Biology, Physiology and Immunology, Universitat Autònoma de Barcelona, 08193 Bellaterra, Barcelona, Spain

**Keywords:** mtDNA, mtRNA, mitochondrial dysfunction, mitochondrial disorders, neurodegeneration, antiviral response, inflammation, innate immunity, interferon

## Abstract

Symbiosis between the mitochondrion and the ancestor of the eukaryotic cell allowed cellular complexity and supported life. Mitochondria have specialized in many key functions ensuring cell homeostasis and survival. Thus, proper communication between mitochondria and cell nucleus is paramount for cellular health. However, due to their archaebacterial origin, mitochondria possess a high immunogenic potential. Indeed, mitochondria have been identified as an intracellular source of molecules that can elicit cellular responses to pathogens. Compromised mitochondrial integrity leads to release of mitochondrial content into the cytosol, which triggers an unwanted cellular immune response. Mitochondrial nucleic acids (mtDNA and mtRNA) can interact with the same cytoplasmic sensors that are specialized in recognizing genetic material from pathogens. High-energy demanding cells, such as neurons, are highly affected by deficits in mitochondrial function. Notably, mitochondrial dysfunction, neurodegeneration, and chronic inflammation are concurrent events in many severe debilitating disorders. Interestingly in this context of pathology, increasing number of studies have detected immune-activating mtDNA and mtRNA that induce an aberrant production of pro-inflammatory cytokines and interferon effectors. Thus, this review provides new insights on mitochondria-driven inflammation as a potential therapeutic target for neurodegenerative and primary mitochondrial diseases.

## 1. Mitochondria in the Eukaryotic Cell: A Double-Edged Sword

Cellular complexity could not have been possible without mitochondria, resulting from the engulfment of a proteobacterium by the precursor of the eukaryotic cell around two billion years ago [1]. Among other acquired key functions, mitochondrial ability to produce ATP through respiration was instrumental to promote eukaryotic development. Conversely, mitochondrial evolution was possible by outsourcing the energetic burden of protein synthesis to the cell by transferring most of their genetic material to the nuclear genome [1,2]. In verterbrates, out of the known ~1500 mitochondrial proteins, only 13 peptides, all involved in the oxidative phosphorylation (OXPHOS) system, are encoded in the mitochondrial circular genome (mtDNA), while the rest are encoded by the nuclear genome and need to be trafficked from the cytoplasm, underscoring the marked dependence of mitochondria on their cellular counterpart [3]. Hence, proper communication between mitochondria and the nucleus—known as retrograde signaling—is essential for cellular function and the survival of the organism [4]. Mutations in both mtDNA and nuclear genes that constitute the mitochondrial proteome are responsible for primary mitochondrial disorders (PMD), a group of highly heterogeneous debilitating human conditions, hallmarked by faulty oxidative phosphorylation (OXPHOS), which affect around 1 in 5000 births [5]. Despite major advances in our understanding of the pathophysiology of PMD, clinical management of these conditions remains largely supportive [5].

Appropriate homeostasis of mitochondria is also essential in the maintenance of cellular health. Thus, many genetic disorders are associated with abnormal mitochondrial function as a secondary phenomenon [6]. Mitochondrial damage can be caused by nongenetic factors, such as mitochondrial toxins, ischemia, infections, or sterile inflammation [7,8,9,10,11,12]. Furthermore, acquired mitochondrial loss-of-function has been linked, with varying levels of evidence, to Parkinson’s disease (PD), Alzheimer’s disease (AD), amyotrophic lateral sclerosis (ALS) [13,14] schizophrenia, and autism [15,16]. Secondary mitochondrial defects are best exemplified by severe, persistent inflammatory states but are also represented in metabolic disorders [17] and some cancers [18].

Mitochondrial dysfunction is commonly associated to high energy-requiring cells, such as neurons. Indeed, neurological damage plays a prominent role in both primary and secondary mitochondrial disease (MD) pathology and lethality. However, not all neuronal populations are equally vulnerable to MD, but rather show a striking anatomical and cellular specificity [19]. Therefore, identifying the underlying mechanisms driving this susceptibility is essential to provide effective therapeutic targets for MD.

It has long been suggested that mitochondria-induced insults, such as proteotoxic or oxidative stress, participate in mitochondrial dysfunction [13,20], but the underlying mechanistic correlates of neuronal vulnerability to mitochondrial dysfunction are far from being elucidated. In this regard, it has become increasingly evident that the endosymbiotic nature of mitochondria poses a pathogenic risk to eukaryotic cells and may contribute to disease progression [21]. In particular, both the mitochondrial genome and its transcriptional products are analogous to their bacterial counterparts. Accordingly, it has been shown that the release of mitochondrial components, such as nucleic acids, into the cytosol, triggers a cell-autonomous innate immune response comparable to those induced by pathogenic bacteria [22,23]. Hence, to limit mitochondria-induced inflammatory damage, bacteria-like cytosolic release of mitochondrial nucleic acids needs to be exquisitely segregated from cytosolic innate response sensors. It is thus noteworthy that PMD patients usually present a worsening of their symptoms after infection, likely linking PMD to antiviral response [24,25,26].

In this review, we discuss current insights regarding the molecular mechanisms of mitochondrial nucleic acid-induced antiviral response with a special focus on how this process contributes to pathogenesis of human diseases that course with mitochondrial dysfunction, inflammation, and neurodegeneration.

## 2. mtDNA and the Immune System: Keeping Your Friend Close and Your Enemy Closer

mtDNA is a clear testament of the bacterial origin of mitochondria. Mammalian mtDNA is a plasmid-like double-stranded circular molecule composed of a heavy and light strand that encode 13 polypeptides of the OXPHOS system, 22 transfer RNAs and 2 ribosomal RNAs (12S and 16S rRNAs) needed for the translation of the mtDNA transcripts [27]. in In the mitochondrial matrix, there are hundreds to thousands copies of mtDNA per cell [28]. Due to its proximity to the electron transport chain, mtDNA is particularly susceptible to oxidation and, therefore, more likely to mutate. In the absence of histones, single molecules of mtDNA are protected from oxidative damage by the transcription factor A (TFAM), which packages them into a slightly elongated, irregularly shaped nucleoid structure [27]. Moreover, consistent with prokaryotic nucleic acid, human mtDNA shows a methylation pattern different to mammalian nuclear DNA, with unmethylated (or hypomethylated) CpG motifs [29,30,31]. Together, these distinctive features make mtDNA an ideal target to be mis-identified as a foreign entity by innate immune sensors. 

## 3. Innate Immune Sensors and mtDNA: To Detect and to “Protect”?

The intracellular surveillance system against pathogen intrusion is driven by multiple pattern-recognition receptors (PRRs) that act as sentinels of foreign nucleic acids and activate intracellular signaling pathways to mount an anti-pathogen immune response [32]. However, PRRs can also interact with endogenous nucleic acids and eventually trigger an indiscriminate defense response against cell’s own components even in a sterile environment [33,34]. In response to cellular stress or mitochondrial insult, the integrity of mitochondrial membranes is compromised and mtDNA is released from mitochondria into the cytosolic compartment. Once in the cytosol, mtDNA can be detected by multiple PRRs, including nucleotide-binding oligomerization domain (NOD)-like receptors (NLRPs), toll-like receptors (TLRs), and the cyclic GMP/AMP synthase–stimulator of interferon genes (cGAS–STING) systems, triggering aberrant pro-inflammatory and type I interferon (IFN) responses [35,36] (Figure 1).

Multiple sensors—Absent in melanoma 2 (AIM2) molecule; NOD, LRR and pyrin domain-containing protein 1 (NLRP1); and NLRP3 or NLR family CARD domain-containing protein 4 (NLRC4)—have been shown to form cytosolic multiprotein complexes, known as inflammasomes, which are activated in response to a broad range of pathogen-derived and endogenous molecules [37,38,39]. Inflammasomes mediate caspase-1 activation and consequent IL-1β and IL-18 maturation, triggering gasdermin D-mediated pyroptotic cell death, an inflammatory form of programmed cell death [38,39] (Figure 1). Inflammasome and caspase-1 activation typically require two signals, namely the priming signal, leading to the upregulation of pro-IL-1β, and a danger signal, promoting inflammasome platform aggregation [38,39]. The assembly of inflammasomes requires a specialized sensor (involved in coupling stimulus recognition to complex assembly), an adaptor molecule, the apoptosis-associated speck-like protein containing a CARD (ASC), and the effector protease caspase-1 [38,39]. 

The connection between mitochondrial dysfunction and inflammasome activation was demonstrated for the first time in macrophages [40,41]. Specifically, Zhou et al. found that mitochondrial reactive oxygen species (ROS) were critical for NLRP3 activation [41], whereas Nakahira et al. proposed that mtDNA was also crucial [40]. These studies showed that mitophagy blockade, a quality control process for the clearance of damaged mitochondria, leads to the accumulation of damaged, ROS-generating mitochondria, resulting in cytosolic translocation of mtDNA and NLRP3 inflammasome activation. NLRP3 in particular has been shown to be preferentially activated by oxidized mtDNA [42]. In fact, it has been recently demonstrated that priming NLRP3 inflammasome with the TLR4 ligand lipopolysaccharide (LPS) promotes new mtDNA synthesis leading to the generation of oxidized mtDNA fragments that are released into the cytosol, inducing NLRP3 activation and enhanced inflammatory responses [43]. Interestingly, in rheumatoid arthritis (RA) T cells, mitochondrial MRE11A, a DNA repair nuclease, has been proposed to protect mtDNA from oxidation and to prevent its cytoplasmic leakage, thus repressing inflammasome activation, cell pyroptosis, and aggressive tissue inflammation [44].

As mentioned above, similar to bacterial DNA, mtDNA contains unmethylated and hypomethylated CpG motifs, which can be recognized by TLR9 in the endolysosomal compartment. The TLR9 signaling pathway results in activation of mitogen-activated protein kinases (MAPK) and NF-κB to trigger pro-inflammatory cytokines production and inflammatory responses [45] (Figure 1). A large body of literature suggests that mtDNA is an endogenous TLR9 agonist [46,47,48,49,50]. For example, mtDNA stress due to ablation of the mitochondrial fusion protein Opa1 in skeletal muscle causes a severe mitochondrial inflammatory myopathy, mediated by the activation of TLR9 by mtDNA [50], which results in the startup of the NF-κB inflammatory program, which contributes to enhanced expression of the hepatokine Fibroblast growth factor 21 (Fgf21) and systemic growth impairment [50]. 

In addition to its role in NLRP3 and TLR9 activation, mtDNA has also been shown to participate in the activation of the cGAS–STING signaling pathway, an important regulator of type I IFN response [22,51,52,53,54,55]. cGAS is a cytosolic DNA sensor that activates innate immune response through the production of the second messenger cyclic GMP–AMP (cGAMP), which is detected by the cyclic-dinucleotide sensor STING [56]. Binding of cGAMP activates STING, which results in the activation of the transcription factors NFκB and the interferon regulatory factor 3 (IRF3) through the kinases IKK and TANK-binding kinase 1 (TBK1), respectively. IRF3 subsequently dimerizes and translocates into the nucleus along with NF-κB to induce the production of type I IFNs and the expression of interferon-stimulated genes (ISG) together with the production of proinflammatory cytokines [56] (Figure 1). In this regard, West et al. showed that mitochondrial stress due to defective packaging of mtDNA into nucleoids upon TFAM depletion results in mtDNA leakage to the cytosol that primes a cGAS–STING-dependent antiviral response [22]. Extending this work, Wu et al. have recently reported that chronic mtDNA stress in *Tfam*^+/−^ mouse embryonic fibroblasts (MEFs) is not associated with NF-κB or interferon gene activation but rather requires the unphosphorylated complex ISGF3 (U-ISGF3), which would induce a sustained activation of a specific class of ISGs including PARP9, an ISG involved in the PARP1-mediated DNA repair pathway [57]. Thus, this form of mtDNA stress would enhance a faster nuclear DNA (nDNA) damage response and a more efficient nDNA repair. Additionally, they showed that in MEFs and cancer cells treated with the anti-tumoral drug doxorubicin not only damages nDNA, but also directly damages mtDNA, resulting in cytoplasmic release and subsequent cGAS–STING-dependent ISG activation, in a similar way to *Tfam* depleted cells [57]. Therefore, mtDNA may act as a genotoxic stress sentinel that mediates a novel mitochondria-to-nucleus stress-signaling pathway to prime nDNA damage and repair responses. Interestingly, persistent mtDNA stress in TFAM-deficient mouse melanoma cells contributes to chemotherapy drug resistance in vitro and in vivo, suggesting that suppression of mtDNA damage and release might be a potential target to prevent cancer chemoresistance [57]. Along the same lines, persistent mtDNA stability in the mutator mice has been shown to mediate aberrant STING-mediated type I IFN activation, leading to metabolic and inflammatory alterations [58]. 

## 4. mtDNA Mitochondrial Escape: A Double-Membrane Prison Break

The mitochondrion, as a semi-autonomous organelle, is separated from the cytoplasm by a double membrane system: the outer mitochondrial membrane (OMM), which delimits the outer contour of mitochondria, and inner mitochondrial membrane (IMM), which delimits the area between the intermembrane space and the matrix. To prevent the unwanted extrusion of mtDNA to the cytosol and its interaction with the innate immune sensors, the permeability of these membranes must be tightly regulated. To date, there are three known main protein complexes involved in the regulation of the permeability of mitochondrial membrane enabling mtDNA release: BAX/BAK and the voltage dependent anion channel (VDAC) oligomers at the OMM and the mitochondrial permeability transition pore (MPTP) at the IMM [59,60,61,62] (Figure 2).

Mitochondrial outer membrane permeabilization (MOMP) following the activation of BAX/BAK is a key event enabling the trafficking of pro-apoptotic molecules from the matrix to the cytosol, initiating the intrinsic apoptotic pathway [63]. Notably, under sub-lethal stress, limited mitochondrial permeabilization (also called minority MOMP) can occur in a reduced subset of mitochondria without triggering cell death. This, in turn, drives DNA-damage and genomic instability, promoting transformation and tumorigenesis [64]. Importantly, MOMP is also crucial for the translocation of mtDNA to the cytosol, which, in the absence of apoptotic caspase activation, is recognized by the cGAS–STING pathway and trigger type I IFN responses and expression of ISGs [59,60]. In addition, caspase-deficient MOMP can also stimulate NF-κB activity through the downregulation of inhibitor of apoptosis proteins exerting the pro-inflammatory role of MOMP besides triggering cell death [65]. Thus, under physiological conditions, caspases are key to enforce the non-immunogenic nature of apoptosis suppressing MOMP-driven inflammatory signaling.

Recent reports have shed light on how matrix-localized mtDNA engages cGAS in the cytoplasm [66,67]. In this regard, it has been shown that BAX/BAK oligomers at the OMM can form extremely large macropores enabling the herniation of the IMM and the access of mitochondrial genome into the cytosol [66,67] (Figure 2). Thus, these studies underscore IMM permeabilization as a hallmark for mtDNA-driven inflammation during MOMP-associated apoptotic cell death. However, it is still controversial whether this process occurs in caspase-deficient conditions or irrespective of caspase activity. Furthermore, differences in the extent of the extruded MIM permeabilization were observed, pointing at cell-type specificity of this process. Interestingly, it has been suggested that mtDNA released from hepatocytes treated with ABT-737, an inhibitor of B-cell lymphoma-extra large (Bcl-xl), is also dependent on BAX/BAK activation and it further accumulates when the lysosomal DNaseII activity is suppressed [68]. Intriguingly, cytosolic mtDNA upon apoptotic stimulation in these cells involves the activation of the TLR9/IFN-β signaling pathway (not the cGAS–STING), leading to induction of RIP1-dependent non-apoptotic cell death [68]. However, the underlying mechanisms driving cell-type specific mtDNA-mediated immune sensors activation remain to be elucidated. 

Besides BAK/BAK oligomerization, mtDNA has been reported to be released from endonuclease G-deficient cells through mitochondrial pores formed by VDAC oligomers [61]. Unlike BAX/BAK macropores, occurring in extreme stress and/or during programmed cell death, the formation of these VDAC oligomers may take place under mild-stress conditions (Figure 2). Direct interaction of amino-terminal amino acids of VDAC1 appears to promote VDAC1 oligomerization and increase mtDNA release. Related to this, mitochondria-associated vaccinia virus-related kinase 2 (VRK2) has been recently highlighted as a new interactor of VDAC1 that would promote the association of mtDNA with VDAC1 as well as VDAC1 oligomerization [69]. Remarkably, the use of VDAC1 oligomerization inhibitor VBIT-4 decreased the release of fragmented oxidized mtDNA and inflammation in a mouse model of systemic lupus erythematosus [61] as well as in amyotrophic lateral sclerosis (ALS) patient iPSc-derived motor neurons, suggesting a potential therapeutic approach for these diseases [51].

Mitochondrial permeability transition pore (mPTP) has also been posited as another mechanism for mtDNA escape [62] (Figure 2). mPTP is a high-conductance nonspecific channel, which opens in response to elevated concentrations of Ca^2+^ and other signals in the mitochondrial matrix. Several mPTP protein components have been proposed, including VDAC at the OMM [70,71] and the ADP/ATP translocase at the IMM [72,73]. Nevertheless, a definitive mPTP composition is still unknown [74,75]. In mammals, mPTP opening is favored by binding of the prolyl cis–trans isomerase cyclophilin D (CyPD), leading to increased IMM permeability and release of matrix components to the cytosol. However, the size of the inner-membrane channel present in mPTP should theoretically preclude mtDNA release. In fact, it has been shown that mtDNA released through this pore is limited to mtDNA fragments of sizes smaller than 700 bp [76]. Interestingly, Nakahira et al. have proposed that NLRP3 inflammasome activation mediates the mPTP formation and subsequent translocation of mtDNA to the cytosol in macrophages exposed to an oxidative insult [40]. Accordingly, pharmacological inhibition of the mPTP with the CyPD-binding drug cyclosporine A (CsA) prevents the activation of the inflammasome and the secretion of IL-1β in this model [40,42]. Inhibition of mPTP formation also abrogated the capacity of the vaccine adjuvant chitosan to promote type I IFN response in dendritic cells through increased mitochondrial stress-mediated mPTP opening, mtDNA release, and cGAS activation [77]. However, it is likely that more than one mechanism contributes to chitosan’s adjuvant activity, as the inflammasome-activating potential of chitosan has been previously demonstrated [78]. Accumulation of *Alu* sequence RNA in retinal pigmented epithelium (RPE) cells, a common feature of age-related macular degeneration, has been associated to the induction of mPTP opening and mtDNA release, which in turn promotes noncanonical-inflammasome activation via engaging the cGAS-driven type I IFNs [79]. Moreover, both pharmacological blockade and genetic deletion of mPTP and VDAC1 in an in vitro model of ALS reduced cytoplasmic accumulation of mtDNA and cGAS–STING activation [51]. These in vitro data demonstrate a mechanism by which TAR DNA binding protein 43 (TDP-43), an RNA/DNA binding protein that accumulates in the cytoplasm of neurons of most patients with sporadic ALS, can mislocalize into mitochondria, opening the mPTP and resulting in a VDAC1-dependent mtDNA leakage, thus linking both protein complexes to the extrusion of mtDNA into the cytosol [51]. Besides the in vitro effect, in vivo administration of CsA has been shown to ameliorate the inflammation associated with thermal injury-induced acute lung injury by reducing the escape of mtDNA and limiting the mtDNA-induced mitochondrial dysfunction in the lung [80].

Altogether, these reports corroborate that inhibition of mitochondrial pores plays an important role in the regulation of mtDNA release and may be a potential therapeutic target in human pathologies in which mtDNA-induced inflammation is involved. Even so, as mentioned before, there are still some gaps remaining regarding the precise mechanisms by which mtDNA is extruded from mitochondria, in particular concerning IMM [66,67]. Moreover, the involvement of additional or alternative mechanisms cannot be ruled out. Indeed, Huang et al. have recently proposed the pore-forming protein Gasdermin D (GSDMD) as another possible pathway for mtDNA release [52]. Activation of caspase-11 by internalized LPS-induced GSDMD cleavage enabling gasdermin-N domain (GSDMD-NT) to translocate and form pores in the mitochondrial membrane, resulting in the release of mtDNA and cGAS–STING pathway activation, inactivation of the cell-cycle regulatory transcription factor YAP1, and impaired transcription of cyclin D genes. Interestingly, these authors showed that deletion of the gene for cGAS in a murine model of inflammatory lung injury restored endothelial regeneration [52]. Hence, preventing GSDMD-induced mitochondrial pore formation and subsequent mtDNA release could serve as a potential strategy to avoid the deleterious proinflammatory effects of mtDNA immune activation.

In summary, these observations highlight that the level of mitochondrial stress is likely to define the MOMP underlying both mtDNA release and the associated type I IFN response (Figure 2). However, the underlying mechanisms driving cell-type specific mtDNA-mediated immune sensors activation remain to be elucidated. 

## 5. Mitochondrial Dysfunction and mtDNA Release: A Crossline between Health and Disease

In the last years, a growing number of studies have suggested that the aberrant activation of the immune response upon mtDNA release may contribute to inflammatory pathology and disease severity of a broad spectrum of human pathologies (some of them already cited before). In this regard, stimulation of the cGAS–cGAMP–STING pathway by cytosolic mtDNA has been shown to play an important role in mediating obesity-induced inflammation and metabolic dysfunction in high-fat diet-induced obese mice [54]. Intriguingly, adipose tissue-specific disruption of disulfide-bond A oxidoreductase-like protein (DsbA-L), a mitochondrial matrix chaperone-like protein with unknown function, impaired mitochondrial function leading to mtDNA leakage and exacerbation of the detrimental inflammatory response [54]. In the same way, mitochondrial dysfunction and mtDNA-cGAS–STING-dependent inflammation has also been associated with the progression of chronic kidney disease, acute kidney injury (AKI), and non-alcoholic steatohepatitis (NASH), among others. Importantly, genetic and pharmacological depletion of the STING markedly attenuated fibrosis and tubular and hepatic inflammation contributing to the amelioration of the disease [53,81]. 

### mtDNA and Neurodegenerative Diseases

Inflammation is a prominent hallmark of neurodegenerative disorders [82]. Hence, mtDNA-driven inflammation can be a critical regulator of disease progression in neurodegenerative processes (Table 1). Therefore, limiting the release of innate immune-activating mtDNA may prove crucial to mitigate inflammation and ameliorate disease progression. In this context, mitophagy has been identified as an important mitochondrial quality control mechanism that prevents mtDNA release by defective mitochondria [83]. This homeostatic process is particularly important in post-mitotic, energetically demanding cells, such as neurons. Accordingly, mitochondrial dysfunction, impaired mitophagy and/or inflammation are common features of several neurodegenerative disorders. Mutations in PINK1, a mitochondrially targeted kinase, or Parkin (encoded by PRKN), a cytosolic ubiquitin ligase, cause familial, autosomal recessive inherited forms of Parkinson’s disease (PD) [84]. PINK1 accumulates on the OMM of damaged mitochondria, activates parkin’s E3 ubiquitin ligase activity, and recruits Parkin to the dysfunctional mitochondrion where it ubiquitinates OMM proteins to instigate their removal by selective autophagy (mitophagy) [84]. A recent study on PINK1/parkin deficient mice have revealed that upon exposure to exhaustive exercise or in the presence of mtDNA mutations, as acute or chronic mitochondrial stresses, the loss of PINK1/parkin cause a STING-meditated type I interferon response, presumably via the release of mtDNA into the cytosol [85].

Consequently, genetic inactivation of STING completely prevented the inflammatory response and the resulting dopaminergic neurodegeneration and motor defects, thus linking mitochondria-dependent aberrant neuroinflammation to PD pathology [85]. Subsequent research by the same group has shown that serum from patients carrying PRKN/PINK1 biallelic mutation had increased levels of pro-inflammatory interleukin-6 (IL6) and circulating cell-free mtDNA pointing at a role of STING-mediated inflammation in the pathogenicity of human PD [90]. Surprisingly, loss of STING, or its downstream effector relish (homologous to NF-κB), was insufficient to suppress the locomotor deficits or mitochondrial disruption in Pink1/parkin mutant flies [91]. Hence, it is likely that the central role of STING in the observed response to PINK/parkin deficiency is a mammal-specific response. 

Consistent with these studies, a recent report has shown that cytosolic mtDNA accumulation due to lysosomal impairment induces cytotoxicity in vitro and PD phenotypes in vivo, leading to type I IFN responses and neurodegeneration [86]. Furthermore, this study corroborates that mtDNA is increased in post-mortem brain tissues from PD patients, supporting a common deleterious role of mtDNA in PD pathogenesis [86]. Thus, therapeutic strategies aimed to reduce cytosolic mtDNA leakage and/or accumulation may represent a very promising approach for PD.

As previously discussed, maintenance of protein homeostasis is pivotal for preservation of functional integrity of neuronal mitochondria. The process involved in the proteolysis of mitochondrial damaged or malfunctioning proteins is known as mitochondrial proteostasis. Failure in this mitochondrial quality control process has also been recently linked to enhanced mtDNA-induced cGAS–STING-mediated antiviral gene expression. Two of the main proteins involved in this process include the serine protease caseinolytic mitochondrial matrix peptidase proteolytic subunit (CLPP) (in the inner matrix) and the i-AAA protease YME1L (IMM-anchored in the intermembrane space). CLPP-knockout mice develop a phenotype reminiscent of Perrault syndrome associated with immune alterations, showing elevated expression of ISGs in peripheral tissues and fibroblasts [92]. Torres-Odio et al. demonstrated that loss of CLPP in human and murine engages IFN-I responses via the cGAS–STING axis [55]. Interestingly, they found that CLPP-null cells exhibit markedly disrupted nucleoid architecture and TFAM aggregation compared with controls, suggesting a new role of CLPP in the regulation of mtDNA maintenance [55]. These results would place mtDNA instability and escape into the cytosol upstream of cGAS–STING–IFN-I signaling in cells lacking CLPP. On the other hand, nervous-system-specific Yme1l knockout mice (NYKO mice) present a neuromuscular phenotype with retinal inflammation as an early event [93]. In this regard, work from Langer’s lab has shown that the innate immune activation in deficient YME1L retinas would be driven by mtDNA release via cGAS–STING–TBK1 pathway. The authors show that YME1L is necessary for efficient de novo pyrimidine metabolism by ensuring their synthesis and by preventing the accumulation of the pyrimidine carrier SLC25A33, thereby limiting mitochondrial pyrimidine uptake. Thus, imbalanced of mitochondrial nucleotide supply due to YMEL1 deficiency would cause mtDNA stress, mtDNA release, and enhanced inflammatory response [94].

In light of these findings, multiple recent studies have associated the activation of the mtDNA-cGAS–STING pathway, or more broadly, aberrant innate immune signaling, with the development of some neurodegenerative pathologies, such as amyotrophic lateral sclerosis (ALS) or Huntington’s disease (HD) [51,87].

Cytosolic accumulation of TDP-43 is a common pathological hallmark of most ALS patients and defines the major pathological subtype of frontotemporal lobar degeneration (FTLD) [95]. When TDP-43 invades mitochondria, it causes mtDNA release followed by the induction of type I interferons and inflammatory cytokines in a cGAS–STING-dependent fashion in ALS models in vitro and in vivo [51]. Supporting the clinical relevance of these findings, postmortem spinal cord samples of patients with sporadic ALS exhibited elevated levels of cGAMP, further indicating mtDNA-mediated cGAS–STING activation as critical determinants of TDP-43-associated inflammation and neuropathology [51]. Notably, STING inhibition in ALS patient iPSC-derived motor neurons and TDP-43^Tg/+^ ALS mouse model ameliorated neurodegeneration and mitigated rapid disease progression, pointing out the translational potential of targeting this pathway in ALS [51]. Accordingly, patients with *C9orf72* gene mutation (the most common genetic cause of ALS and FTLD) showed an elevated type I interferon signature and inflammatory response that was mitigated upon STING blockade [96].

Moreover, upregulation of cGAS-STING expression has also been observed in the striatum of a mouse model and HD human patient’s tissue [87,88]. Interestingly, administration of the antioxidant hormone melatonin reduced mitochondrial ROS damage and the consequent mtDNA release, thus ameliorating the cGAS-mediated neuroinflammation and increasing survival of a R6/2 mouse model of HD [87]. Given the fact that melatonin progressively decreases with human aging and in neurodegenerative diseases, and that melatonin deficiency induces cytosolic accumulation of mtDNA and activation of the cGAS pathologic inflammatory response, this protective effect may not be just limited to HD. Thus, these results underscore a marked correlation between mtDNA accumulation and inflammation in the context of neurodegenerative diseases. Interestingly, a recent report from the West lab, using a model of mtDNA instability (the mutator mouse), has identified alterations in the nuclear erythroid 2–related factor 2 (NRF2)-mediated antioxidant transcriptional landscape as a key event for mtDNA-mediated immune responses [58]. The authors show that mtDNA instability and release in POLG mutator mice engages an aberrant STING-dependent IFN-I response, which blocks the activity of NRF2, increases oxidative stress, and contributes to the hyperinflammatory status and progeroid phenotype found in these mice [58]. Mitochondrial dysfunction and chronic inflammation are key features of senescence and neurodegeneration [97,98], and imbalance of the mitochondrial quality control axis has also been described during aging and neurodegenerative disorders [99]. Thus, it is feasible that mtDNA and IFN-I dysregulation may be key initiators in age-related pathology in humans.

## 6. Mitochondrial RNA: A New Villain in Town

In addition to mtDNA, mitochondrial RNA (mtRNA) has recently emerged as a new intracellular alarm signal that can be mistaken as foreign by the innate immune sensors [23,89] (Figure 3). Mitochondria is a key element in the cellular control of viral infections thanks to the mitochondrial localization of the antiviral-signaling protein (MAVS) as a hub for viral RNA-mediated responses [100,101]. Analogously, it has been shown that mtRNA released in the cytoplasm is recognized by the antiviral RNA receptors RNA helicase retinoic acid inducible protein I (RIG-I) and melanoma differentiation associated gene 5 (MDA5), which activate the mitochondrial adaptor MAVS via CARD–CARD interactions promoting MAVS oligomerization to form prion-like structures aggregates [23,100,101,102]. MAVS then complexes the cytoplasmic TBK1 and IKB kinase-ε (IKKε) resulting in activation of the transcription factors interferon regulatory factors 3 and 7 (IRF3/7) and NFκB [100,101,102]. This process ultimately results in the upregulation of the expression of type I IFN, ISGs, and proinflammatory cytokines. 

Therefore, in absence of a viral infection, mtRNA can initiate an antiviral signaling pathway and activate MAVS by an autocrine-like mode creating a feedforward loop that may lead to a sustained immune response and compromised cellular survivor. In particular, double-stranded mitochondrial RNA (mtdsRNA) species are strong activators of the antiviral immune response [23,103]. Owing to their bacterial origin, bidirectional transcription of the light (L) and heavy (H) strands of the circular mtDNA produces near-genome-length polycistronic transcripts that encompass all of the coding information on each strand [27]. These overlapping transcripts are able to form long double-stranded RNA structures, reminiscent of viral replication products, hence potentially highly immunogenic [23,104]. In this sense, the mtRNA-decay machinery, the so-called degradosome, plays an important role in maintaining mitochondrial integrity, mediating mtRNA surveillance and degradation, especially of the non-coding L-strand transcripts. The mitochondrial degradosome complex is composed of the ATP-dependent RNA and DNA helicase SUV3 and the polynucleotide phosphorylase PNPase (encoded by *PNPT1*), which has poly(A) polymerase (PAP) and exoribonuclease activities [105]. Notably, depletion of either enzyme results in massive accumulation of mtdsRNA [23,106]. Intriguingly, only PNPase but not SUV3 blockade results in mtdsRNA mislocalization into the cytosol where it engages an MDA5-driven type I IFN response through the MAVS signaling pathway [23]. Moreover, silencing of BAX/BAK after PNPase dysfunction prevented *IFNB1* mRNA induction suggesting that mtdsRNA escape depends on BAX/BAK pores [23]. Importantly, fibroblasts from four different patients with hypomorphic mutations in *PNTP1* displayed mtdsRNA accumulation as well as upregulation of ISGs and other markers of immune activation in peripheral blood samples [23]. These results highlight a crucial role of PNPase in preventing the formation and release of mtdsRNA into the cytoplasm and the induction of an inappropriate innate immune response with deleterious pathological consequences.

In keeping with this, cytosolic mtdsRNA also induces type I IFN expression via the MDA5–MAVS pathway in several cancer cell lines. Interestingly, this response can be reverted under hypoxia, likely by downregulation of mtDNA transcription [107]. Hence, mtdsRNA-induced type I IFN expression may not only depend on intrinsic defects of the mitochondrial degradosome but can be also influenced by the metabolic status of the cell. 

It is also noteworthy that in a parallel study, Kim et al. reported that mtdsRNA represent the majority of endogenous molecules that bind the dsRNA sensor protein kinase RNA-activated (PKR) [103]. Activated PKR undergoes auto-phosphorylation (pPKR) and suppresses global translation by phosphorylating eukaryotic translation initiation factor 2 alpha (peIF2α) on Ser51. Additionally, decreasing the amount of mtRNAs by knocking down POLRMT, a mitochondrial RNA polymerase, resulted in decreased levels of pPKR and peIF2α [103]. Intriguingly, authors posit that PKR-mtdsRNA interaction may occur inside mitochondria. Subsequently, PKR needs to be translocated to the cytosol to induce its downstream signaling cascade. However, the underlying mechanisms governing how PKR may be able to move in and out of mitochondria remains unclear. These results suggest that PKR activation by mtdsRNAs is a dynamically regulated process, which can be counteracted by PKR phosphatases PP1 and PP2A. During M phase of the cell cycle, endogenous activation of PKR by mtdsRNA can affect the translational status of the cell. However, overactivation of PKR due to severe stress conditions can trigger cell death. Remarkably, this study proposes the mtdsRNA–PKR interaction as crucial signaling axis for the regulation of cellular metabolism [103]. 

Supporting this new role of PKR as mediator in the mitochondria-cytosol crosstalk, a recent study reported that mutant huntingtin-induced mitochondrial dysfunction leads to mtRNA leakage and activation of PKR, inducing innate immune activation [89]. Accordingly, upregulation of mtRNA was found in post-mortem brains from Huntington disease (HD) patients and HD mouse models correlating with disease progression [89]. Interestingly, this response was selectively restricted to striatal spiny projection neurons (SPN) from human patients, which are the most affected neurons by HD, suggesting a potential key role of mtRNA-induced immune response in the increased vulnerability of these neurons [89]. Along the same lines, in vitro chemical inhibition of the mitochondrial electron transport chain (ETC) in wild-type iPCS-derived human neurons is sufficient to induce PKR expression [89]. Together, even though the mechanistic insight is still missing, these studies pave the way for further assessment of the relation between mitochondrial dysfunction and innate immune response, which may contribute to identify the molecular determinants of the cell-type specific vulnerability in neurodegeneration in the quest to find novel therapeutic targets. 

In keeping with this, Tigano et al. recently proposed a new regulatory function for mtRNAs release as an intrinsic mechanism of immune surveillance involved in cellular sensing of mtDNA damage [108]. The study described a novel mechanism by which mtDNA breaks promote the formation of the BAX/BAK pore and the escape of mtRNA into the cytoplasm followed by activation of the RIG-I-MAVS signaling pathway and upregulation of effectors of the innate immune response [108]. Consistently with this finding, an analogue mechanism in corneal keratocytes after surgery or trauma and leading to blindness has been proposed [109]. As a consequence of their exposition to UV radiation, keratocytes in the central cornea accumulate mtDNA deletions that affect the ETC function and increase oxidative stress leading to the release of dsmtRNA [109]. Subsequent cytoplasmic dsmtRNA activates the MDA5-NF-κB molecular cascade, exacerbating the production of the proinflammatory cytokine IL-8, which alters the wound healing process causing vision loss [109]. Thus, these studies highlight the crucial role of mitochondrial nucleic acids altering the retrograde signaling between mitochondria and nucleus and its potential implication in the pathogenesis of inflammatory and mitochondrial disorders.

## 7. Concluding Remarks: PAMP-Ing the Cell up

Self-preservation is an axiom in biology. As such, all life forms, ranging from unicellular to complex organisms, have developed mechanisms to ensure survival. Thus, cellular defense mechanisms in front of external pathogens are paramount. In this respect, the appearance of innate immune pathways in metazoans has allowed the development of systems to identify, isolate, and counteract pathogens such as bacteria and viruses, even at the expense of the demise of the infected cell. However, cellular death can be extremely detrimental in the case of post-mitotic cells, such as neurons. Hence, it is critical to avoid aberrant activation of these anti-pathogen systems. In this regard, the origin and evolution of mitochondria has placed this organelle in a unique position to mediate cellular antipathogen responses. Accordingly, mitochondria have long been identified as a key mediator of cellular survival and death and, more recently, as a critical integrator of the cellular response to external pathogens, such as viruses. 

In addition to these known activities, a growing wealth of evidence is underscoring the role of mitochondria as a trigger of a host-derived aberrant antiviral response via the release of mitochondrial nucleic acids, leading to cellular shutdown and death. The particularities of the bacterial-like mitochondrial DNA and mitochondrial translation make them able to activate virtually all cellular pathogen sensors and have several critical implications. Firstly, on a strictly theoretical level, it begs the question whether mt nucleic acids should be considered damage-associated molecular patterns (DAMPS) or rather pathogen-associated molecular patterns (PAMPS). Even though mitochondria have co-evolved with eukaryotic cells for billions of years, the fact that mitochondrial components are still identified as pathogens seems to underscore that the mitochondrion is, in fact, a trojan horse inside the cell, albeit a very useful one. Furthermore, on a more pathologically-relevant note, the correlation between mitochondrial (dys)function, antiviral responses, and neurodegeneration highlights the importance of proper mitochondrial activity to cellular survival. It is noteworthy that cellular heterogeneity in mitochondrial function, mt-derived DAMPs (or PAMPs) regulation, or activation threshold of antiviral responses may govern cellular fate. Hence, mt-induced antiviral responses may underlie the marked cell-type specificity observed in diseases with mitochondrial involvement. In addition, activation of type I IFN signaling by mt-derived nucleic acids may lead to the propagation of an inflammatory response, which would further contribute to secondary damage and neurodegeneration (Figure 4, Table 1).

However, many open questions remain to fully understand the functional relevance of this pathway. Cellular dissection of the cell types implicated in the initiation of this response, as well as identification of the factors involved in nucleic acid release, and a possible crosstalk between intracellular sensors will provide highly valuable insights on the contribution of mitochondria-triggered neuronal death and open new avenues for novel therapeutic approaches for mitochondrial (and likely neurodegenerative) diseases.

## Figures and Tables

**Figure 1 ijms-22-08523-f001:**
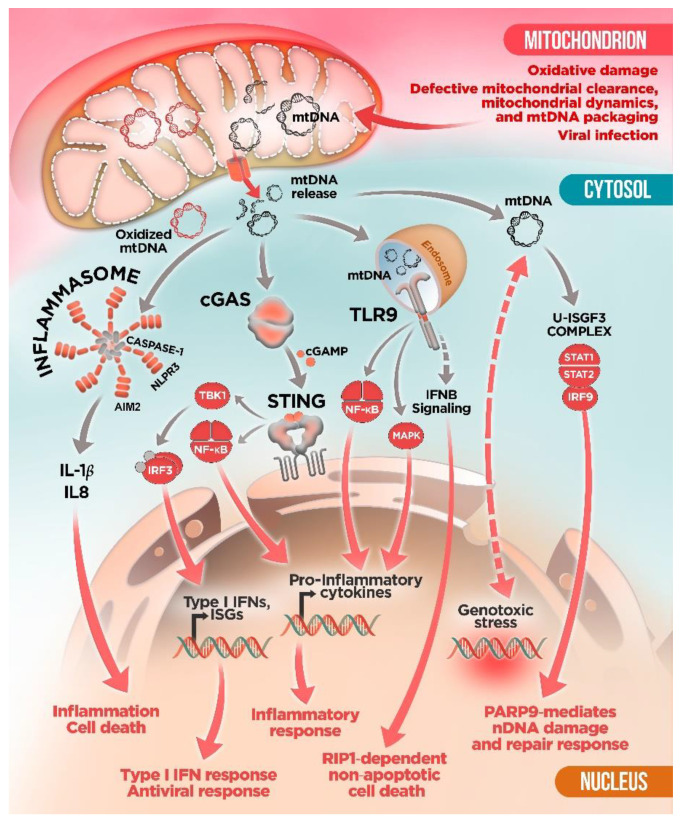
mtDNA-activated immune pathways. Under mitochondrial insults (oxidative stress, infections, etc.) or mitochondrial quality control impairment, mtDNA is released to the cytosol where it engages different receptors, activating inflammatory and innate immune signaling pathways. From the left: Oxidized mtDNA stimulates inflammasome activity leading to caspase-1-mediated maturation of pro-inflammatory IL-1β and IL18; mtDNA activates the cGAS–STING pathway triggering interferon response via IRF3 and NF-κB; TLR9 recognizes mtDNA, resulting in activation of NF-κB or MAPK and pro-inflammatory cytokines production. mtDNA also promotes non-apoptotic cell death trough TLR9/IFN-β signaling. Genotoxic stress and DNA repair response are also mediated by cytosolic mtDNA.

**Figure 2 ijms-22-08523-f002:**
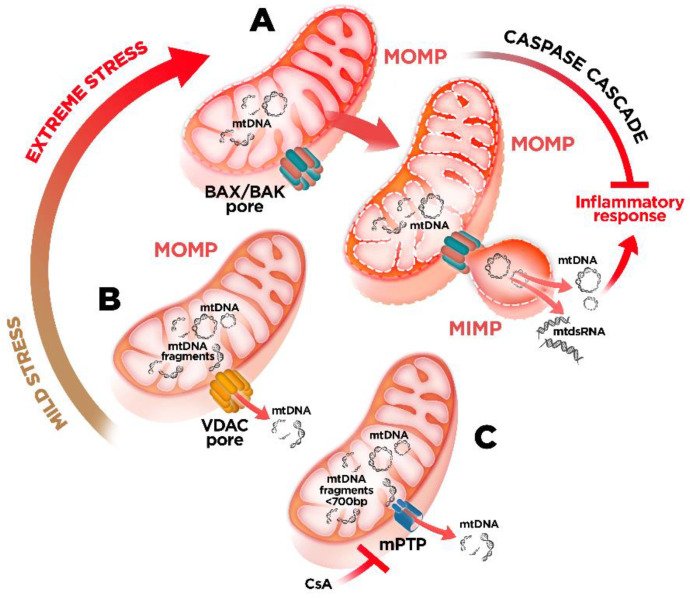
Proposed mechanisms of mtDNA release to the cytosol: (**A**) Under extreme stress conditions, activation of BAX/BAK leads to MOMP. Following MOMP, the outer membrane pores progressively widen, allowing for the extrusion of inner mitochondrial membrane into the cytosol and its permeabilization to release mitochondrial nucleic acids. (**B**) Under mild-stress conditions, VDAC oligomer pores promote MOMP and thus make mtDNA release into the cytosol possible. (**C**) mPTP opening allows the release of small mtDNA fragment. Abbreviations: MOMP: mitochondrial outer membrane permeabilization; MIMP: mitochondrial inner membrane permeabilization; MPTP: mitochondrial permeability transition pore; CsA: cyclosporine A.

**Figure 3 ijms-22-08523-f003:**
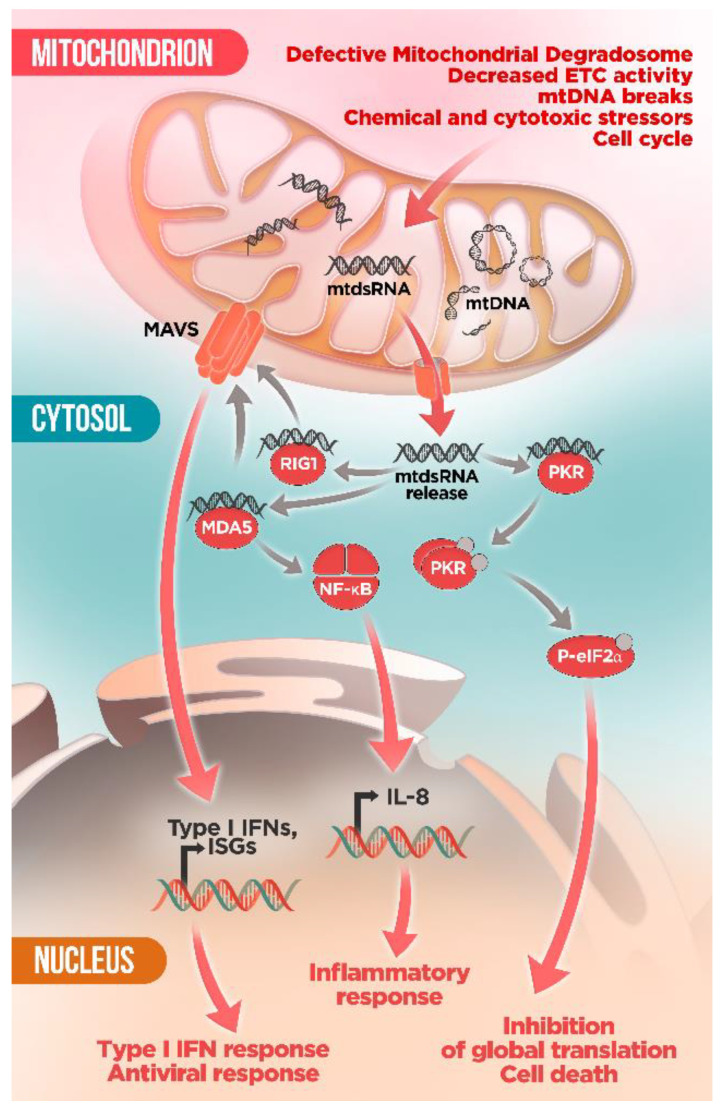
mtdsRNA-activated immune pathways. Dysfunctional mitochondria can accumulate mtdsRNA intermediates, either as a consequence of mitochondrial defects or due to external stimuli. Once released into the cytosol, mtdsRNA binds to the immune sensors MDA5 and RIG1, eliciting interferon type I response via MAVS. mtdsRNA can also bind and activate the antiviral kinase PKR, leading to phosphorylation of eIF2α and inhibition of global translation or cell death under sustained PKR activation.

**Figure 4 ijms-22-08523-f004:**
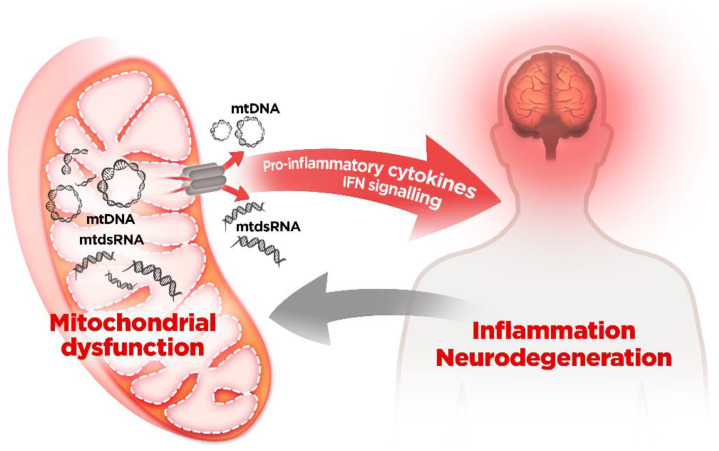
Mitochondrial nucleic acids-driven inflammation as a critical regulator of disease progression in neurodegenerative processes. Dysregulation of mitochondrial function leads to the activation of type I IFN signaling by mt-derived nucleic acids. The activation and the propagation of the inflammatory response may contribute to secondary damage and neurodegeneration. Besides mitochondrial dysfunction promoting inflammation, inflammation per se can lead to mitochondrial dysfunction, suggesting the existence of a pro-inflammatory loop with mitochondria as a central player.

**Table 1 ijms-22-08523-t001:** Role of mtDNA and mtRNA in neurodegenerative diseases.

Pathology	Mitochondial Nucleic Acid Involvement
Parkinson’s Disease (PD)	Cytosolic mtDNA accumulation associated with type I IFN responses and PD phenotypes in vitro and in vivo [85,86]. Circulating cell-free mtDNA and increased mtDNA in postmortem brain tissues from PD patients [86].
Perrault syndrome	mtDNA instability and escape into the cytosol drives cGAS–STING–IFN-I signaling in cells lacking CLPP [55].
Amyotrophic lateral sclerosis (ALS)	mtDNA-mediated cGAS–STING pathway activation leads to induction of type I IFNs and inflammatory cytokines in TDP-43 accumulating cells [51].
Huntington’s disease (HD)	Upregulation of cGAS–STING expression in the striatum of a mouse model and HD human patient’s tissue [87,88].mtDNA release in R6/2 mouse model of HD [87].Upregulation of mtRNA in post-mortem brains from HD patients and mouse models [89].
Leigh Syndrome	mtdsRNA accumulation in fibroblasts from patients *PNTP1* Mutations and upregulation of ISGs and markers of immune response in peripheral blood [23].

## Data Availability

Not applicable.

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
