# Peer review of "Mitochondria-Induced Immune Response as a Trigger for Neurodegeneration: A Pathogen from Within"

_ijms, 2021, doi:10.3390/ijms22168523_

Round 1

Reviewer 1 Report

The review by Luna-Sanchez and colleagues provides an account of the role mitochondria play in immune responses. Mitochondria lie at the crossroads of many signaling cascades and current research is revealing that both mtDNA and mtRNA are immunogenic antigens. The review discusses how mtDNA may promote immune activation. It then describes mechanisms whereby mtDNA may escape the bilayer membranes of the mitochondria to initiate the immune responses in the cytosol, an important unsolved issue in this line of research. The review then discusses mtDNA in the context of neurodegenerative diseases. Finally, the review explores the newer findings of how mtRNA may also contribute to immune system activation. This is a timely review, as there is great interest in this topic and many outstanding questions are yet to be solved in this field.

The title suggests ‘neuronal immune response’ but the review is neither focused on neuronal tissues or responses but cites more broad literature. I would suggest the authors consider revising the title to reflect this.

Line 88 It really is a misconception to suggest mtDNA lack protection because they lack protective histones, even though the statement is factually true. As the authors rightly point out, mtDNA is wrapped and protected by TFAM. Please revise.

Line 42 The authors make the statement “Mutations in both mtDNA and nuclear genes that constitute the mitochondrial respiratory chain (MRC), are responsible for primary mitochondrial disorders (PMD), a group of highly heterogeneous invalidating human conditions, hallmarked by faulty oxidative phosphorylation (OXPHOS) that affect >1 in 5000 birth [5].” Not just the genes encoding the MRC components are the reason for primary mitochondrial disorders, please revise to include the fact that genes encoding Pol gamma, Twinkle and other such metabolic enzymes also give rise to PMD.

Line 116 Figure 1, why not include AIM2 along with NLPR3 inflammasome in figure 1 since it’s the first thing discussed immediately following the graphic.

The review needs a comprehensive review for verb-noun agreements, many small typographical errors, and none-standard English phrases (Invalidating disorder, mitochondrial affection, mtDNA-medianted).

Author Response

We would like to thank the reviewer for the interest in the manuscript and for the insightful comments. We have addressed his/her concerns as described below in the point by point responses:

The title suggests ‘neuronal immune response’ but the review is neither focused on neuronal tissues or responses but cites more broad literature. I would suggest the authors consider revising the title to reflect this.

We agree with reviewer that the mechanisms described have been observed in different cell types and models. In part, this is due to the fact that this field is a relatively new one in the context of neural tissue. However, we believe that its relevance in neuronal death and neurodegeneration is a focus of our review. Thus, we have revised the title for accuracy and clarity, as suggested.

Line 88 It really is a misconception to suggest mtDNA lack protection because they lack protective histones, even though the statement is factually true. As the authors rightly point out, mtDNA is wrapped and protected by TFAM. Please revise.

Reviewer is correct and we agree the statement may be confusing. We have revised the text accordingly.

Line 42 The authors make the statement “Mutations in both mtDNA and nuclear genes that constitute the mitochondrial respiratory chain (MRC), are responsible for primary mitochondrial disorders (PMD), a group of highly heterogeneous invalidating human conditions, hallmarked by faulty oxidative phosphorylation (OXPHOS) that affect >1 in 5000 birth [5].” Not just the genes encoding the MRC components are the reason for primary mitochondrial disorders, please revise to include the fact that genes encoding Pol gamma, Twinkle and other such metabolic enzymes also give rise to PMD.

We thank reviewer for pointing out this issue. We have modified the text as suggested.

Line 116 Figure 1, why not include AIM2 along with NLPR3 inflammasome in figure 1 since it’s the first thing discussed immediately following the graphic.

We have now included AIM2 in the figure, as suggested.

The review needs a comprehensive review for verb-noun agreements, many small typographical errors, and none-standard English phrases (Invalidating disorder, mitochondrial affection, mtDNA-medianted).

We apologize for the tyopographical errors ans incorrect sentences. We have edited the text accordingly.

Reviewer 2 Report

The present study describes the involvment of mitochondria in innate immunity at many levels. Loss of mitochondrial function or more in general multiple step pathways of mitochondrial qualitycan trigger inflammation. Mitochondria share molecular patterns with batcteri and viruses that may contribute to inflammatory events underlying several disorders including neurodegeneration. Several mitochondrial-activated immune pathways are reported also discussing the presence of targets with possible therapeutical implications.

Complessively the paper is very rich and detailed and show a complete review on the topic. 

Athough the information provided are logically and well organized, in my opinion (but is only a suggestion not a request) it would be worth to summarise the potential terapeutic approaches, with special concern to neuroinflammation, in a dedicated paragraph.

The only criticism refers to the omission of a proper deepenig on the connection between the accumulation of mtDNA mutations and increased inflammation during aging.  

Author Response

We would like to thank reviewer 2 for his/her enthusiasm on this review. A point-by-point response is detailed below:

Athough the information provided are logically and well organized, in my opinion (but is only a suggestion not a request) it would be worth to summarise the potential terapeutic approaches, with special concern to neuroinflammation, in a dedicated paragraph.

We appreciate the suggestion of the reviewer and while we agree it will be informative to describe potential therapeutic approaches, we feel that at this point they would be either too speculative  at the moment or already covered by excellent existing reviews.

The only criticism refers to the omission of a proper deepenig on the connection between the accumulation of mtDNA mutations and increased inflammation during aging.  

Following reviewer's suggestion we are now including a brief discussion on its potential role on aging (line 417).

Reviewer 3 Report

The manuscript by by Luna-Sanchez et al. is a very interesting review about the role of damage mitochondria  in  innate immunity. 

Minor revision:

To improve the readability of the text, I would suggest introducing a table with neurodegenerative diseases in which mtDNA plays a role. I would also add a list that combines the abbreviations used in the text (not only in the figures' captions).

Author Response

We thank reviewer for the positive feedback on our work. In response to reveiewer's comments, we have added a new Table (Table 1) describing the neurodegenerative diseases with mtDNA involvement as well as a final abbreviations list.